# Mental health and burnout during medical school: Longitudinal evolution and covariates

**Valerie Carrard**[1], **Sylvie Berney**[2], **Céline Bourquin**[1], **Setareh Ranjbar**[3], **Enrique Castelao**[3], **Katja Schlegel**[4], **Jacques Gaume**[5], **Pierre-Alexandre Bart**[6], **Marianne Schmid Mast**[7], **Martin Preisig**[3], **Alexandre Berney**[1] *

1 Department of Psychiatry, Psychiatric Liaison Service, Lausanne University Hospital (CHUV) and University of Lausanne, Lausanne, Switzerland, 2 Department of Psychiatry, Service of General Psychiatry, Lausanne University Hospital (CHUV) and University of Lausanne, Lausanne, Switzerland, 3 Department of Psychiatry, Psychiatric Epidemiology and Psychopathology Research Centre, Lausanne University Hospital (CHUV) and University of Lausanne, Lausanne, Switzerland, 4 Institute of Psychology, University of Bern, Bern, Switzerland, 5 Department of Psychiatry, Addiction Medicine, Lausanne University Hospital (CHUV) and University of Lausanne, Lausanne, Switzerland, 6 Department of Internal Medicine, Lausanne University Hospital (CHUV) and University of Lausanne, Lausanne, Switzerland, 7 Faculty of Business and Economics (HEC Lausanne), Department of Organizational Behavior, University of Lausanne, Lausanne, Switzerland

* alexandre.berney@chuv.ch

## Abstract

### Background

Medical students' rate of depression, suicidal ideation, anxiety, and burnout have been shown to be higher than those of the same-age general population. However, longitudinal studies spanning the whole course of medical school are scarce and present contradictory findings. This study aims to analyze the longitudinal evolution of mental health and burnout from the first to the last year of medical school using a wide range of indicators. Moreover, biopsychosocial covariates that can influence this evolution are explored.

### Method

In an open cohort study design, 3066 annual questionnaires were filled in by 1595 different students from the first to the sixth year of the Lausanne Medical School (Switzerland). Depression symptoms, suicidal ideation, anxiety symptoms, stress, and burnout were measured along with biopsychosocial covariates. The longitudinal evolution of mental health and burnout and the impact of covariates were modelled with linear mixed models.

### Results

Comparison to a same-aged general population sample shows that medical students reported significantly more depression symptoms and anxiety symptoms. Medical students' mental health improved during the course of the studies in terms of depression symptoms, suicidal ideation, and stress, although suicidal ideation increased again in the last year and anxiety symptoms remained stable. Conversely, the results regarding burnout globally showed a significant worsening from beginning to end of medical school. The covariates

**Data Availability Statement:** The data that support the main findings of this study are openly available in zenodo at http://doi.org/10.5281/zenodo. 8405764. For the comparison analyses, the data of

CoLaus|PsyCoLaus study used in this article cannot be fully shared as they contain potentially sensitive personal information on participants. According to the Ethics Committee for Research of the Canton of Vaud, sharing these data would be a violation of the Swiss legislation with respect to privacy protection. However, coded individual-level data that do not allow researchers to identify participants are available upon request to researchers who meet the criteria for data sharing of the CoLaus|PsyCoLaus Datacenter (CHUV, Lausanne, Switzerland). Any researcher affiliated to a public or private research institution who complies with the CoLaus|PsyCoLaus standards can submit a research application to research.colaus@chuv.ch or research.psycolaus@chuv.ch. Proposals requiring baseline data only, will be evaluated by the baseline (local) Scientific Committee (SC) of the CoLaus and PsyCoLaus studies. Proposals requiring follow-up data will be evaluated by the follow-up (multicentric) SC of the CoLaus|PsyCoLaus cohort study. Detailed instructions for gaining access to the CoLaus|PsyCoLaus data used in this study are available at www.colaus-psycolaus.ch/professionals/how-to-collaborate.

**Funding:** This work was supported by the Swiss National Science Foundation (grant number 10001C_197442). The CoLaus|PsyCoLaus offspring study was supported by unrestricted research grants from GlaxoSmithKline, the Faculty of Biology and Medicine of Lausanne, the Swiss National Science Foundation (grants 3200B0–105993, 3200B0-118308, 33CSCO-122661, 33CS30-139468, 33CS30-148401, 33CS30_177535, 3247730_204523, 324730_189130) and the Swiss Personalized Health Network (grant 2018DRI01). The funders had no role in study design, data collection and analysis, decision to publish, or preparation of the manuscript.

**Competing interests:** The authors have declared that no competing interests exist.

most strongly related to better mental health and less burnout were less emotion-focused coping, more social support, and more satisfaction with health.

## Conclusion

Both improvement of mental health and worsening of burnout were observed during the course of medical school. This underlines that the beginning and the end of medical school bring specific challenges with the first years' stressors negatively impacting mental health and the last year's difficulties negatively impacting burnout.

## Introduction

Medical students have been shown to present more depression [1], anxiety [2], and burnout [3] than same-aged general population individuals. Meta-analyses have indeed reported a pooled prevalence of 28.0% of the medical students for depression [1,4], 11.1% for suicidal ideation [1], 33.8% for anxiety [2], and 44.2% for burnout [5]. Moreover, a study has shown that before actually starting medical school, medical students present less depression and less burnout than same-aged general population; suggesting that the course of the medical school itself may trigger mental health issues and burnout [6].

Despite the wealth of literature on the subject, longitudinal studies analyzing the evolution of mental health and burnout from beginning to end of medical school are still scarce and present inconsistent findings. For depression symptoms, a meta-analysis estimated a median increase of 13.5% each curriculum year [1], but another meta-analysis reported a gradual decrease until the last year of medical school [4]. For anxiety, both increase and decrease trends have been reported [7] and for suicidal ideation and burnout, longitudinal studies are too scarce to provide a reliable understanding of their evolution during medical school [1,8]. Consequently, the patterns of evolution of mental health and burnout during medical school are still unknown.

Medical students' mental health issues and burnout have been attributed to different factors. Entering medical school represent a significant source of stress as it often marks a transition of moving away from home towards a new social environment with new ways of studying that demands an important adjustment from students [4]. Then, academic factors of stress are numerous such as the volume of information to learn, time constraints, examinations, competition, ethical challenges, and exposure to human suffering [9]. On the other hand, protective factors have been identified as well: a systematic review reports that the factors most often related to less mental health issues and burnout in medical students are low neuroticism, less emotion-focused coping strategies, more physical activity, lower study-related stress, and higher satisfaction with studies [10].

The present study aims to add to the existing literature by analyzing the temporal evolution of mental health and burnout throughout the course of medical school [1,8]. The few studies using prospective data focused mostly on the first year of medical school or on the two first years [1,8]. Yet, it seems essential to understand the temporal evolution of medical students' mental health and burnout throughout the entire medical school curriculum in order to identify the curriculum years that are most challenging. Moreover, there is a need for more studies analyzing conjointly a wide range of biopsychosocial covariates of medical students' mental health and burnout in order to identify the most influential protective and aggravating factors. The objectives of the present study are thus to (1) compare the mental health and coping

strategies of medical students to same-age general population, (2) describe the evolution of mental health and burnout from beginning to end of medical school using a wide range of indicators (depression symptoms, suicidal ideation, anxiety symptoms, stress, emotional exhaustion, cynicism, and academic efficacy), and (3) identify biopsychosocial covariates influencing the evolutions of medical students' mental health and burnout.

## Method

### Design and data collection

This is a longitudinal analysis of data from the ETMED-L project [11] collected with an open cohort study design. An online questionnaire investigating mental health and interpersonal competence was sent to all medical students (curriculum years 1 to 6) matriculated at the University of Lausanne (Switzerland), except for the students who are at the university as part of an academic exchange. It took approximately one hour to fully complete the online questionnaire and the students received 50CHF (~50USD) for each completed questionnaire. The present study used the data of the three first waves of the ETMED-L project collected during the months of March 2021, November 2021, and November 2022.

To compare the mental health of medical students to same-aged general population individuals, data collected as part of the CoLaus|PsyCoLaus were retrieved. Initiated in 2003, CoLaus|PsyCoLaus is a population-based cohort study that follows a representative sample of the residents of Lausanne (Switzerland). The study has been designed to investigate psychiatric disorders and cardiovascular risk factors as well as their association using structured interviews and questionnaires [12]. For the present study, the data of the offspring of the participants of CoLaus|PsyCoLaus collected from the 26[th] of March 2008 to the 16[th] of November 2022 were used.

The Research Ethics Committee of the Canton de Vaud approved both the ETMED-L (project number 2020–02474) and CoLaus|PsyCoLaus (project number PB_2018–00038, reference 239/09) projects. All participants gave written informed consent.

### Measures

**Mental health and burnout.** The instruments used in the ETMED-L project are detailed in its published protocol [11]. For the present paper, four measures of ***mental health*** were used. *Depression symptoms* were assessed with the Center for Epidemiological Studies-Depression [CES-D; 13] with the validated cut-offs scores of 16 for males, 20 for females, and 19 overall [14]. *Suicidal ideation* was measured with two questions of the Beck Depression Inventory [BDI; 15]. *Anxiety symptoms* were assessed with the State-Trait Anxiety Inventory [STAI; 16] and stress was measured with a single generic item ("Globally, how would you evaluate your current stress level on a scale from 1 'none' to 10 'extreme'?"). For ***burnout***, the Maslach Burnout Inventory Student-Survey [MBI-SS; 17] was used. The Maslach Burnout Inventory is the most widely used measure of burnout that assesses three dimensions: emotional exhaustion, depersonalization, and personal accomplishment (reversed dimension). The students' version used in the present study has been validated among students and measures the corresponding three dimensions of *Emotional Exhaustion*, *Cynicism*, and *Academic Efficacy* (reversed dimension). Except for the single item measuring stress, all instruments are well established and have a French validated version that has been used in the present study.

**Biopsychosocial covariates.** Four ***sociological covariates*** were analyzed: identifying as male (gender identity recoded into 1 = male and 2 = female or non-binary), paid job (hours per week), parents' education (number of parents with college or university degree), having a partner (1 = yes, 0 = no), and social support. Social support was measured as the average score

of two questions adapted from a larger cohort study's questionnaire [18]: "If necessary, in your opinion, to what extent can someone provide you with practical help, this means concrete help or useful advice, if 0 means "not at all" and 10 "a great deal"?" for practical support and "To what extent can someone be available in case of need and show understanding, by talking with you for example, if 0 means "not at all" and 10 "a great deal"?" for emotional support (Cronbach's alpha = .73).

Regarding *psychological covariates*, having consulted a psychotherapist during the past year (1 = yes, 0 = no) and coping strategies were assessed. Coping strategies were measured with the French version of the coping section of the Euronet questionnaire [19,20]. Confirmatory factor analysis [21] supported three factors including the dimensions of *emotion-focused*, *help-seeking*, and *problem-focused coping strategies*.

Finally, three *biological covariates* were included: Body Mass Index (BMI; weight in kilograms divided by height in meters squared), physical activity (number of hours per week), and satisfaction with health ("Are you satisfied with your health?" rated on a scale from 1 = very unsatisfied to 5 = very satisfied).

## Statistical analysis

Medical students were compared to same-age general population individuals using linear regression with sample (medical students vs general population) as independent variable, being male as a control variable, and depression symptoms, anxiety symptoms, and coping strategies (the only indicators measured similarly in both the ETMED-L and CoLaus|PsyCo-Laus projects) separately as dependent variables. Cohen's *d* was computed for effect size estimation with *d*s of 0.2, 0.5, and 0.8 respectively considered as small, medium, and large effect sizes [22]. For a more clinically relevant appreciation of the results, a test of proportion was additionally used to compare the two samples in terms of risk of clinical depression according to the CES-D cutoff. Note that for all these comparison analyses, medical students' data at first participation were used.

Linear Mixed Models (LMMs) were used to model the longitudinal evolution of mental health and burnout over the course of medical school. Due to funding constraints, the students could not be followed during the entire six-year of medical school. However, LMMs provides a powerful framework for modeling the longitudinal evolution of mental health and burnout across the entire six-year duration of medical school, even when the medical students have been assessed at a maximum of three time points. This approach indeed effectively accounts for limited and unevenly spaced data points, facilitating insights into trajectories over the entire course of medical school.

The LMMs analyses followed the subsequent steps for each mental health and burnout indicators separately (see S1 File for the related equations). In a first step, Model 1 was fitted considering the student specific effects as random intercepts to account for the clustering in the data due to the repeated measures for each medical student and including all the biopsychosocial covariates described in the Measure section. In a second step, two different temporal variance-covariance structures were fitted to potentially account for the temporal spillover of mental health and burnout (i.e. the influence of past observation on current observation). The same model as in the first step was fitted, where an Autoregressive Covariance Structure of AR (1) was considered in Model 2 and an Autoregressive/Moving Average Covariance Structure of ARMA(1,1) in Model 3. Then, the best fitting model between 1, 2, and 3 was selected using likelihood ratio tests. In a third step, the non-linear evolution of mental health and burnout was tested by including the time as quadratic in Model 4 and cubic in Model 5 with the temporal structure selected in the second step. Using likelihood ratio tests, Models 4 and 5 were then

compared to the linear Model 1, 2, or 3 according to the temporal structure selected in the second step. The best fitting model of this third step was presented as the final model. Note that Restricted Maximum Likelihood method was used to produce unbiased estimates of variance and covariance parameters [23,24]. For all the final models, we report marginal $R^2$ as the percentage of variation explained by the fixed part of the model, conditional $R^2$ as the percentage of variation explained by both fixed and random part of the model, and Intraclass Correlation Coefficient (ICC) as the proportion of total between-student variance. Moreover, Standardized β are presented for effect sizes estimation. Standardized βs between 0.10–0.29 are considered as small, between 0.30–0.49 as medium, and 0.50 or greater as large effect sizes [22].

The highest missing rate at the item level was 0.97% (coping and social support items). It has been shown that it is unlikely to be much gain from multiple imputation when missing rates are lower than 5% [25]. Thus, missing data at the item level were replaced by mean scores if less than 20% of the items were missing. If 20% or more of the items were missing, the total score was considered missing and missing data at the score level were then handled with full information maximum likelihood in the LMMs. Stata version 17 [26] and R version 4.2.2 [27] were used for the analyses and p values < .05 were considered significant.

## Results

### Samples

The eligible students who agreed to participate were 906, 1059, and 1101 for each data collection wave, resulting in a total of 3066 questionnaires (a detailed participation flow chart is provided as S1 Fig). From those, 74 questionnaires were excluded because students gave a wrong answer to at least one of the two attention questions placed in the questionnaire (e.g., "In order to check your attention, please answer 'Slightly agree' to this question.") and 6 questionnaires were excluded due to technical issues. Thus, each data collection wave included respectively 886, 1034, and 1066 answered questionnaire, which represent 49.41%, 49.33%, and 54.92% of each wave's eligible students. Then, 168 students who repeated a year had to be excluded to have only linear curriculum trajectories. The final sample thus included 2601 questionnaires form 1595 different students with 909 students who participated in only one wave, 366 who participated in two waves, and 320 who participated in three waves of data collection.

The same-age CoLaus|PsyCoLaus offspring comparison group, included 91 participants who completed at least one of the questionnaires of interest and gave consent for the reuse of their individual data. They were between 18 and 35 years old with a mean age of 22.99 (SD = 4.31) and 60.92% were women.

### Descriptive statistics

The medical student sample descriptive is displayed in Table 1. Medical students were between 17 and 49 years old with a mean age of 21.78 (SD = 3.14) and most of them self-identified as females (67.90%), which corresponds to the gender proportions generally observed in the Lausanne Medical School. Observed means and standard deviation of medical students' mental health and burnout for each curriculum year are detailed in Table 2 and correlations between the variables of interest are provided as (S1 Table).

### Comparison to same-age general population

Results of linear regressions testing the difference between medical students and same-age general population individuals with respect to depression symptoms, anxiety symptoms, and coping strategies are displayed in Table 3. Results indicated that medical students presented

**Table 1. Sample statistics as indicated at first participation (N = 1595).**

|  | %Missing | N | % | M | SD | Min | Max |
|---|---|---|---|---|---|---|---|
| **Age** | 0.19 | 1592 |  | 21.78 | 3.14 | 17 | 49 |
| **Gender identity** | 0.00 | 1595 |  |  |  |  |  |
| Male |  | 499 | 31.29 |  |  |  |  |
| Female |  | 1083 | 67.90 |  |  |  |  |
| Non-Binary |  | 13 | 0.82 |  |  |  |  |
| **Paid job (hours/week)** | 0.13 | 1593 |  | 2.67 | 5.98 | 0 | 50 |
| **Parents education** | 0.00 | 1595 |  |  |  |  |  |
| None with higher education |  | 407 | 25.52 |  |  |  |  |
| One with higher education |  | 346 | 21.69 |  |  |  |  |
| Both with higher education |  | 842 | 52.79 |  |  |  |  |
| **Having a partner** | 0.13 | 833 | 52.23 |  |  |  |  |
| **Social support** | 0.56 | 1586 |  | 7.95 | 1.98 | 0 | 10 |
| **Having consulted a psychotherapist** | 0.19 | 387 | 24.26 |  |  |  |  |
| **Coping** |  |  |  |  |  |  |  |
| Emotion-focused | 0.50 | 1587 |  | 9.82 | 4.05 | 0 | 24 |
| Problem-focused | 0.50 | 1587 |  | 7.33 | 1.78 | 1 | 12 |
| Help-seeking | 0.56 | 1586 |  | 5.47 | 2.89 | 0 | 12 |
| **BMI** | 0.31 | 1590 |  | 21.78 | 3.03 | 15.64 | 41.67 |
| **Physical activity (hours/week)** | 0.19 | 1592 |  | 3.11 | 2.70 | 0 | 21 |
| **Satisfaction with health** | 0.19 | 1592 |  | 3.75 | 1.01 | 1 | 5 |

*Note.* Higher education of parents means a college or university degree.

significatively more depression symptoms and anxiety symptoms than the same-age general population. A test of proportion further showed that the proportion of individuals who were likely to be affected with clinical depression according to the CES-D cutoffs were significantly higher in the medical student sample (46.88%) than in the same-age general population sample (21.69%), Z = 4.49, p < .001. Regarding coping, medical students reported using significantly less help-seeking strategies than same-age general population but did not differ in terms of emotion-focused nor problem-focused coping strategies. Cohen's *d*s show that the difference in depression symptoms was of medium effect size and that it was small for anxiety symptoms and help-seeking coping strategies.

## Longitudinal evolution of mental health and burnout

Results of the LMMs modelling the longitudinal evolution of mental health during medical school are displayed in Table 4. They showed a significant linear decrease of depression symptoms and stress from the first to the sixth year of medical school. For suicidal ideation, the quadratic function of curriculum years was significant, indicating that evolution was convex (U shape), with a decrease in suicidal ideation until the third year of medical school followed by an increase until the end of the curriculum. No significant change over curriculum years was found for anxiety symptoms.

The results for burnout's longitudinal evolution during medical school are displayed in Table 5. They show that academic efficacy (reversed dimension of burnout) significantly decreased linearly from the first to the last year of medical school. For cynicism, we observed a slow increase in the two first years of medical school followed by a gradually steeper increase in the last years (i.e., a significant convex course following a sort of J shape). Finally, a

**Table 2. Descriptive statistics of students' mental health and burnout (N = 1595).**

| | N | %Missing | Mean | SD | Min | Max | Skewness | Kurtosis | Cronbach's α |
|---|---|---|---|---|---|---|---|---|---|
| **Mental Health** | | | | | | | | | |
| *Depression symptoms* (Center for Epidemiological Studies-Depression [CES-D]) | | | | | | | | | |
| Year 1 | 592 | .84 | 21.95 | 11.62 | 0 | 55 | 0.49 | 2.68 | .92 |
| Year 2 | 358 | .56 | 19.91 | 10.71 | 0 | 50 | 0.53 | 2.66 | .91 |
| Year 3 | 466 | .21 | 18.38 | 11.22 | 0 | 58 | 0.81 | 3.31 | .93 |
| Year 4 | 404 | .49 | 16.53 | 10.11 | 0 | 52 | 0.79 | 3.37 | .91 |
| Year 5 | 425 | .23 | 15.93 | 10.71 | 0 | 56 | 0.96 | 3.87 | .93 |
| Year 6 | 343 | .58 | 14.96 | 10.73 | 0 | 51 | 0.84 | 3.19 | .93 |
| *Suicidal Ideation* (two items of the Beck Depression Inventory [BDI]) | | | | | | | | | |
| Year 1 | 592 | .84 | 0.86 | 1.12 | 0 | 6 | 1.35 | 4.46 | .47 |
| Year 2 | 358 | .56 | 0.63 | 0.97 | 0 | 5 | 1.66 | 5.39 | .37 |
| Year 3 | 466 | .21 | 0.58 | 1.03 | 0 | 6 | 2.10 | 7.61 | .54 |
| Year 4 | 404 | .49 | 0.55 | 0.98 | 0 | 5 | 1.93 | 6.25 | .50 |
| Year 5 | 425 | .23 | 0.63 | 0.99 | 0 | 5 | 1.59 | 4.94 | .44 |
| Year 6 | 343 | .58 | 0.59 | 0.95 | 0 | 4 | 1.64 | 5.11 | .40 |
| *Anxiety symptoms* (trait section of the State-Trait Anxiety Inventory [STAI]) | | | | | | | | | |
| Year 1 | 592 | .84 | 46.90 | 12.15 | 20 | 78 | 0.03 | 2.42 | .93 |
| Year 2 | 358 | .56 | 45.00 | 11.61 | 20 | 76 | 0.16 | 2.57 | .93 |
| Year 3 | 466 | .21 | 44.18 | 11.46 | 20 | 80 | 0.07 | 2.53 | .93 |
| Year 4 | 404 | .49 | 43.38 | 11.71 | 20 | 78 | 0.18 | 2.36 | .93 |
| Year 5 | 425 | .23 | 42.48 | 11.77 | 20 | 74 | 0.13 | 2.26 | .93 |
| Year 6 | 342 | .87 | 42.12 | 11.85 | 20 | 73 | 0.17 | 2.30 | .94 |
| *Stress* (single generic item) | | | | | | | | | |
| Year 1 | 592 | .84 | 5.86 | 2.09 | 1 | 10 | -0.49 | 2.23 | . |
| Year 2 | 358 | .56 | 5.65 | 2.16 | 1 | 10 | -0.28 | 1.96 | . |
| Year 3 | 466 | .21 | 5.38 | 2.02 | 1 | 9 | -0.22 | 2.03 | . |
| Year 4 | 404 | .49 | 5.21 | 2.16 | 1 | 10 | -0.15 | 1.85 | . |
| Year 5 | 425 | .23 | 5.36 | 2.20 | 1 | 10 | -0.21 | 2.00 | . |
| Year 6 | 342 | .87 | 4.62 | 2.31 | 1 | 10 | 0.25 | 1.88 | . |
| **Burnout** (Maslach Burnout Inventory Student-Survey [MBI-SS]) | | | | | | | | | |
| *Emotional Exhaustion* | | | | | | | | | |
| Year 1 | 592 | .84 | 17.19 | 5.00 | 5 | 30 | 0.19 | 2.68 | .85 |
| Year 2 | 358 | .56 | 17.84 | 4.88 | 5 | 30 | -0.01 | 2.68 | .86 |
| Year 3 | 466 | .21 | 17.19 | 4.83 | 6 | 30 | 0.30 | 2.90 | .86 |
| Year 4 | 404 | .49 | 16.35 | 4.80 | 5 | 30 | 0.33 | 3.05 | .86 |
| Year 5 | 425 | .23 | 15.60 | 5.13 | 5 | 30 | 0.43 | 2.95 | .88 |
| Year 6 | 342 | .87 | 15.32 | 5.16 | 5 | 30 | 0.19 | 2.63 | .88 |
| *Cynicism* | | | | | | | | | |
| Year 1 | 592 | .84 | 9.18 | 4.47 | 4 | 24 | 0.98 | 3.56 | .84 |
| Year 2 | 358 | .56 | 8.88 | 3.77 | 4 | 24 | 0.87 | 3.58 | .79 |
| Year 3 | 466 | .21 | 8.98 | 4.07 | 4 | 24 | 1.10 | 4.11 | .83 |
| Year 4 | 404 | .49 | 9.75 | 4.59 | 4 | 24 | 0.96 | 3.53 | .88 |
| Year 5 | 425 | .23 | 10.59 | 4.81 | 4 | 24 | 0.71 | 2.88 | .88 |
| Year 6 | 342 | .87 | 11.02 | 4.91 | 4 | 24 | 0.63 | 2.71 | .88 |
| *Academic Efficacy* | | | | | | | | | |
| Year 1 | 592 | .84 | 24.04 | 4.56 | 9 | 36 | -0.17 | 2.91 | .75 |
| Year 2 | 358 | .56 | 24.56 | 4.18 | 14 | 36 | -0.03 | 2.73 | .71 |

*(Continued)*

**Table 2.** (Continued)

| | N | %Missing | Mean | SD | Min | Max | Skewness | Kurtosis | Cronbach's α |
|---|---|---|---|---|---|---|---|---|---|
| Year 3 | 466 | .21 | 24.46 | 4.32 | 11 | 36 | -0.24 | 2.81 | .76 |
| Year 4 | 404 | .49 | 24.07 | 4.56 | 11 | 36 | -0.11 | 2.87 | .77 |
| Year 5 | 425 | .23 | 24.01 | 4.89 | 6 | 36 | -0.34 | 3.05 | .80 |
| Year 6 | 342 | .87 | 24.23 | 4.77 | 10 | 35 | -0.38 | 3.08 | .79 |

*Note*. Skewness > 2 and kurtosis > 7 indicate substantial non-normality [28].

curvilinear evolution was observed (~ shape) for the emotional exhaustion dimension with an increase in emotional exhaustion until the second year of medical school, followed by a decrease until the fifth year, and finally another increase until the sixth year.

## Influence of biopsychosocial covariates

The results for the covariates' influence are displayed in Tables 4 and 5. The biopsychosocial factors that were significantly related to all outcome variables were gender identification, social support, having consulted a psychotherapist during the last year, emotion-focused coping, and satisfaction with health. Social support and satisfaction with health were consistently related to *less* mental health issues and burnout, whereas having consulted a psychotherapist during the last year and emotion-focused coping were consistently related to *more* mental issues and burnout. For gender, identifying as male was significantly associated with less depression symptoms, anxiety symptoms, stress, and emotional exhaustion, but it was related to more suicidal ideation, more cynicism, and less academic efficacy (reversed dimension of burnout).

**Table 3. Comparison between medical students and same-age general population of mental health and coping at first participation.**

| Variables | N | M | SD | t | Cohen's d | F | R² |
|---|---|---|---|---|---|---|---|
| **Depression symptoms** | | | | 4.61*** | 0.51 | 54.95*** | 0.07 |
| Medical students | 1587 | 19.00 | 11.33 | | | | |
| General population | 83 | 12.24 | 9.61 | | | | |
| **Anxiety symptoms** | | | | 2.15* | 0.25 | 46.96*** | 0.07 |
| Medical students | 1587 | 44.57 | 12.08 | | | | |
| General population | 79 | 40.70 | 11.78 | | | | |
| **Emotion-focused coping** | | | | 0.53 | 0.06 | 75.89*** | 0.12 |
| Medical students | 1587 | 9.82 | 4.05 | | | | |
| General population | 79 | 9.67 | 3.81 | | | | |
| **Problem-focused coping** | | | | 0.30 | 0.04 | 2.88* | 0.00 |
| Medical students | 1587 | 7.33 | 1.78 | | | | |
| General population | 79 | 7.43 | 1.56 | | | | |
| **Help-seeking coping** | | | | 2.15* | 0.25 | 30.40*** | 0.04 |
| Medical students | 1586 | 5.47 | 2.89 | | | | |
| General population | 78 | 6.18 | 2.81 | | | | |

*p < .05

***p < .001.

*Note*. Linear regression models including being male and age as a control variable were used. Depression symptoms was measured with the Center for Epidemiological Studies-Depression (CES-D), anxiety symptoms with the trait section of the State-Trait Anxiety Inventory (STAI), and coping strategies with the Euronet questionnaire.

**Table 4. Linear mixed models of the evolution of mental health including biopsychosocial covariates.**

| | Depression Symptoms | | | Suicidal Ideation | | | Anxiety symptoms | | | Stress | | |
|---|---|---|---|---|---|---|---|---|---|---|---|---|
| | Std β | B | CI | Std β | B | CI | Std β | B | CI | Std β | B | CI |
| (Intercept) | .01 | 29.21*** | 25.32 – 33.09 | -.05 | 1.48*** | 1.04 – 1.92 | .01 | 46.92*** | 42.88 – 50.97 | .00 | 6.35*** | 5.46 – 7.25 |
| Curriculum year | -.11 | -0.69*** | -0.89 – -0.49 | .00 | -0.14** | -0.23 – -0.04 | .00 | -0.03 | -0.24 – 0.18 | -.10 | -0.12*** | -0.17 – -0.08 |
| Identifying as male | -.06 | -1.37** | -2.24 – -0.50 | .05 | 0.12* | 0.02 – 0.21 | -.08 | -2.12*** | -3.06 – -1.18 | -.06 | -0.29** | -0.49 – -0.09 |
| Paid job | .01 | 0.02 | -0.03 – 0.07 | .02 | 0.00 | -0.00 – 0.01 | -.01 | -0.03 | -0.08 – 0.02 | -.01 | 0.00 | -0.01 – 0.01 |
| Parents education | .01 | 0.11 | -0.33 – 0.54 | .01 | 0.01 | -0.04 – 0.06 | -.02 | -0.23 | -0.69 – 0.24 | -.05 | -0.14** | -0.23 – -0.04 |
| Having a partner | .00 | 0.06 | -0.63 – 0.75 | -.03 | -0.06 | -0.14 – 0.01 | -.02 | -0.39 | -1.09 – 0.32 | .04 | 0.19* | 0.03 – 0.34 |
| Social support | -.16 | -0.95** | -1.14 – -0.76 | -.11 | -0.06*** | -0.08 – -0.04 | -.11 | -0.71*** | -0.90 – -0.51 | -.07 | -0.08*** | -0.12 – -0.03 |
| Psychotherapist consultation | .12 | 3.26*** | 2.46 – 4.07 | .06 | 0.13** | 0.04 – 0.22 | .15 | 4.32*** | 3.50 – 5.13 | .07 | 0.37*** | 0.19 – 0.55 |
| Emotion-focused coping | .40 | 1.14*** | 1.05 – 1.24 | .32 | 0.08*** | 0.07 – 0.09 | .44 | 1.33*** | 1.23 – 1.43 | .30 | 0.16*** | 0.14 – 0.19 |
| Problem-focused coping | .01 | 0.07 | -0.11 – 0.26 | -.01 | -0.01 | -0.03 – 0.02 | -.02 | -0.16 | -0.35 – 0.02 | .01 | 0.01 | -0.03 – 0.06 |
| Help-seeking coping | -.09 | -0.34** | -0.48 – -0.21 | -.09 | -0.03*** | -0.05 – -0.02 | -.08 | -0.32*** | -0.45 – -0.18 | .00 | 0.00 | -0.03 – 0.03 |
| BMI | -.03 | -0.10 | -0.22 – 0.02 | -.02 | -0.01 | -0.02 – 0.01 | .00 | -0.01 | -0.14 – 0.13 | .00 | 0.00 | -0.03 – 0.03 |
| Physical activity | -.04 | -0.18** | -0.31 – -0.05 | .00 | 0.00 | -0.01 – 0.02 | -.03 | -0.13 | -0.27 – 0.00 | -.05 | -0.04** | -0.07 – -0.01 |
| Satisfaction with health | -.19 | -2.22*** | -2.56 – -1.88 | -.17 | -0.18*** | -0.22 – -0.14 | -.13 | -1.53*** | -1.87 – -1.19 | -.16 | -0.35*** | -0.43 – -0.27 |
| Curriculum year^2 | | | | .06 | 0.02** | 0.01 – 0.03 | | | | | | |
| $\sigma^2$ | 44.24 | | | 0.50 | | | 35.25 | | | 2.25 | | |
| $\tau_{00 \text{ students}}$ | 23.75 | | | 0.32 | | | 38.93 | | | 1.32 | | |
| ICC | .35 | | | .39 | | | .52 | | | .37 | | |
| N $_{students}$ | 1584 | | | 1584 | | | 1584 | | | 1584 | | |
| N $_{observations}$ | 2583 | | | 2583 | | | 2583 | | | 2583 | | |
| Marginal $R^2$ | .44 | | | .22 | | | .43 | | | .23 | | |
| Conditional $R^2$ | .64 | | | .52 | | | .73 | | | .52 | | |
| AICc | 18129.18 | | | 6760.85 | | | 18143.15 | | | 10539.82 | | |

*p < .05

**p < .01

***p < .001.

*Note*. Std = Standardized. Depression symptoms was measured with the Center for Epidemiological Studies-Depression (CES-D), suicidal ideation with two questions of the Beck Depression Inventory (BDI), anxiety symptoms with the trait section of the State-Trait Anxiety Inventory (STAI), and stress with a single generic item ("Globally, how would you evaluate your current stress level on a scale from 1 'none' to 10 'extreme'?").

Other covariates were related to fewer mental health or burnout variables. Help-seeking coping was associated to less mental health issues (on all mental health variables), but unrelated to burnout. Physical activity was related to less depression symptoms, stress, and emotional exhaustion. Higher parent education was related to less stress, but more cynicism. Having a partner was related to less stress and emotional exhaustion. And finally, problem-focused coping was related to more academic efficacy. Having a paid job and BMI were the only tested covariates that showed no significant influence.

According to effect size indicators, the strongest predictors of students' mental health and burnout were emotion-focused coping (average absolute β = .33), followed by satisfaction with health (average absolute β = .14) and social support (average absolute β = .11). The effect sizes are considered medium for emotion-focused coping and small for all other significant covariates. Overall, the models tested explained between 52% (for suicidal ideation) and 73% (for anxiety symptoms) of the outcome variables' total variance (fixed effect plus random structure; "Conditional $R^2$").

**Table 5. Linear mixed models of the evolution of burnout including biopsychosocial covariates.**

| | Emotional Exhaustion | | | Cynicism | | | Academic Efficacy | | |
|---|---|---|---|---|---|---|---|---|---|
| | *Std β* | *B* | *CI* | *Std β* | *B* | *CI* | *Std β* | *B* | *CI* |
| (Intercept) | .07 | 12.69*** | 10.35 – 15.02 | -.04 | 9.63*** | 7.63 – 11.63 | -.01 | 19.69*** | 17.83 – 21.55 |
| Curriculum year | -.33 | 4.68*** | 3.31 – 6.05 | .17 | -0.03 | -0.43 – 0.37 | -.10 | -0.25*** | -0.35 – -0.16 |
| Identifying as male | -.04 | -0.48* | -0.95 – -0.01 | .05 | 0.50* | 0.04 – 0.96 | -.05 | -0.53* | -0.96 – -0.09 |
| Paid job | .00 | 0.00 | -0.02 – 0.03 | .01 | 0.01 | -0.02 – 0.03 | .00 | 0.00 | -0.02 – 0.02 |
| Parents education | .03 | 0.17 | -0.07 – 0.40 | .05 | 0.26* | 0.03 – 0.48 | -.03 | -0.14 | -0.36 – 0.07 |
| Having a partner | .04 | 0.42* | 0.07 – 0.77 | .00 | 0.01 | -0.33 – 0.34 | .03 | 0.26 | -0.06 – 0.58 |
| Social support | -.08 | -0.21*** | -0.31 – -0.12 | -.10 | -0.23*** | -0.32 – -0.14 | .14 | 0.34*** | 0.25 – 0.42 |
| Psychotherapist consultation | .06 | 0.68** | 0.28 – 1.09 | .07 | 0.69*** | 0.31 – 1.08 | -.08 | -0.90*** | -1.27 – -0.53 |
| Emotion-focused coping | .33 | 0.42*** | 0.37 – 0.47 | .24 | 0.28*** | 0.23 – 0.32 | -.25 | -0.30*** | -0.34 – -0.25 |
| Problem-focused coping | .00 | 0.01 | -0.08 – 0.10 | -.01 | -0.02 | -0.11 – 0.06 | .14 | 0.36*** | 0.27 – 0.44 |
| Help-seeking coping | -.03 | -0.05 | -0.12 – 0.02 | -.01 | -0.01 | -0.08 – 0.05 | .07 | 0.12*** | 0.05 – 0.18 |
| BMI | .03 | 0.05 | -0.01 – 0.12 | -.02 | -0.02 | -0.09 – 0.04 | .01 | 0.02 | -0.04 – 0.08 |
| Physical activity | -.04 | -0.07* | -0.14 – -0.01 | .00 | 0.00 | -0.06 – 0.07 | .01 | 0.02 | -0.04 – 0.09 |
| Satisfaction with health | -.14 | -0.72*** | -0.89 – -0.56 | -.09 | -0.41*** | -0.57 – -0.25 | .12 | 0.56*** | 0.40 – 0.71 |
| Curriculum year^2 | -.09 | -1.56*** | -2.01 – -1.12 | .05 | 0.07* | 0.02 – 0.13 | | | |
| Curriculum year^3 | .15 | 0.14*** | 0.10 – 0.18 | | | | | | |
| $\sigma^2$ | 8.35 | | | 8.61 | | | 7.92 | | |
| $\tau_{00 \text{ students}}$ | 10.30 | | | 8.82 | | | 7.92 | | |
| ICC | .55 | | | .51 | | | .50 | | |
| N $_{\text{students}}$ | 1584 | | | 1584 | | | 1584 | | |
| N $_{\text{observations}}$ | 2583 | | | 2583 | | | 2538 | | |
| Marginal $R^2$ | .24 | | | .12 | | | .20 | | |
| Conditional $R^2$ | .66 | | | .57 | | | .60 | | |
| AICc | 14564.06 | | | 14255.51 | | | 14023.67 | | |

*p < .05

**p < .01

***p < .001.

*Note*. Std = Standardized. Burnout was measured with the Maslach Burnout Inventory Student-Survey (MBI-SS).

## Discussion

The results of the present study show that medical students' mental health gradually improves in terms of depression symptoms, suicidal ideation, and stress, although suicidal ideation increases again during the last year of medical school and anxiety symptoms remain stable throughout the curriculum. For burnout, it is another picture: it worsens with medical students experiencing gradually more cynicism and less academic efficacy, whereas we observe a worsening of emotional exhaustion until the end of the second year, followed by an improvement, and then another surge during the last year.

Our results are in line with past findings regarding the comparison between medical students and general population in terms of mental health [1,2]. This study indeed indicates that medical students report significantly more depression symptoms and anxiety symptoms compared to same-age general population individuals. In fact, according to the CES-D cutoffs, twice as much medical students are at risk of clinical depression compared to same-age general population individuals. Moreover, a specificity/sensitivity check of the CES-D cutoffs using data collected as part of the CoLaus|PsyCoLaus project showed that one third of the individuals who scored above the CES-D cutoffs presented a clinical depression diagnosed with the

Diagnostic Interview for Genetic Studies (a clinical interview assessing major mood and psychotic disorders [29]). Based on these analyses, it can be assumed that around 15% of our participating medical students and up to 20% of the first-year students presented a clinical depression.

Like other studies in the field [4], our results confirm that students present high mental health issues in the first two years of medical school. On the bright side, our study indicates like other past research [4] that medical students' mental health mostly improved along the curriculum years. Nevertheless, the present study indicates that the students presented a surge in suicidal ideation and burnout in the last year of medical school. Given that few longitudinal studies span over the whole medical curriculum, the difficulties met in the last year of medical school has seldomly been described in the past. The stressors faced at the end of medical school are manifestly different from the ones medical students face in the first years of their studies. Indeed, the beginning of medical school is characterized by difficulties that might be faced by any new university students, such as change of social environment, adaptation to a new learning style, high study workload, strong competition, and stressful examinations. The impact of these stressors is likely to lessen along the curriculum years as students adapt to what has become their new lifestyle. As proposed by the Stress and Coping Model [30], medical students' mental health might improve as they progressively appraise their new situation as less challenging and acquire new coping strategies to manage the stressors encountered. However, the end of medical school can bring new types of stressors that might be more specific to medical students. Indeed, through clerkships, medical students are gradually exposed to their potential future practice. They experience the relationship with patients, but also with coworkers and their hierarchy and might face difficult situations where they are confronted with the complexity, feeling of powerlessness, and lack of control that can characterize medical practice. Based on the Job Demand Control Model (JDCM [31]), such an increase in job demands (e.g., more responsibilities and workload [32]) coupled with perceived loss of control (e.g., role conflicts and limit of their theoretical knowledge [32]) can lead to increased stress and strain reactions, including burnout. As they enter clinical practice, the students face their shortcomings and the need to lean on their elders, yet asking for help is a coping mechanism they employ to a limited degree compared to the general population, as our results show. Moreover, the perspective of the transition to the upcoming internship phase can be an important preoccupation for medical students. The end of studies, anticipated as the long-awaited start of professional practice also corresponds to the start of postgraduate training. This period of transition brings forth both choices to make and challenges to overcome. The results of the present study indicate that these stressors faced at the end of medical school impact mental health and burnout differently than the ones faced in the beginning of medical school. In the past decade, there has been a debate on whether burnout and mental health indicators such as depression and anxiety are the same or different constructs [33]. This study indicates that mental health indicators and burnout are related but distinct. Indeed, it shows that mental health is more impacted by the generic stressors of the beginning of medical training, whereas burnout is more impacted by the specific stressors of the end of medical school, which might be characterized by occupational problematics. On the other hand, suicidal ideation and the emotional exhaustion dimension of burnout were influenced by both the stressors of the beginning and the end of medical school, which is in line with previous findings showing that suicidal ideation correlates with both depression and burnout [34] and that emotional exhaustion is the burnout dimension most strongly correlated to depression [35].

The present study replicates most of the gender differences usually observed in the general population [36] with students identifying as male presenting less mental health issues than students identifying as female or non-binary. For burnout, our study also aligns with past

literature showing inconsistent gender differences [37–40]. We indeed found that students identifying as male present less emotional exhaustion, but more cynicism, and less academic efficacy (reversed dimension of burnout) than students identifying as female or non-binary. Gender differences in mental health and wellbeing have been attributed to multiple factors including differences between women and men in the stressors faced, the coping strategies used, and developed vulnerabilities [36]. Indeed, the few studies testing gender differences in the stressors faced by medical students indicate that female medical students are more affected by specific stressors such as the heavy workload of the studies [41], the interactions with teaching staff [42], or the exposure to human suffering [43]. Regarding coping resources, studies showed that female medical students use significantly more emotional support and instrumental support than male students, whereas male students use significantly more humor as coping mechanism [44]. More studies on the gender differences in medical students' stressors and resources and their impact are needed to potentially propose avenues for gender-specific interventions.

This study further shows that the most influential protective factors against medical students' mental health issues and burnout are less emotion-focused coping, more social support, and more satisfaction with health. These covariates were already known to contribute to medical students' mental health and burnout [10] and can be the target of specific student support services. For instance, increasing the use of help-seeking coping strategies has the potential to bring at the same time more adapted coping strategies and more social support. Our results show relatively low help-seeking coping strategies among the medical students compared to same-age general population and past studies report on medical students' reluctance to seek help for mental health issues or burnout because they felt a pressure to be seen as a competent, feared stigmatization, and had insufficient knowledge of existing support services [45]. In our university, numerous extra-curricular activities are proposed by different associations to bring students together and thus increase their social support resources. The problem that might face medical students especially is the difficulty to find room in their busy schedule for such activities, which clearly questions the structure of the curriculum. In addition, medical students are somewhat isolated from the rest of the university campus. Other services available in our university are psychotherapeutic consultations on campus. However, these structures are still limited in their means in comparison to the needs and our results pleads for an increase in the dedicated resources. Emotion-focused coping, defined as the detrimental use of rumination, negative emotions, denial, and giving up, is the most influential among the covariates tested. It thus seems essential to warrant accessible psychotherapeutic help, which has the potential to impact medical students' coping strategies and in turn their mental health [46,47].

## Strengths and limitations

The present study's strength lies in its longitudinal design covering the sixth years of Medical School as well as a large sample. Additionally, the wide array of mental health and burnout validated indicators as well as the large coverage of biopsychosocial potential covariates ensures a more comprehensive picture of the topic. Nevertheless, some limitation needs to be acknowledged. First, a yearly measurement might fail to capture short-term changes in mental health and burnout. Second, the results for stress are to be interpreted with caution given that it was measured with a single item that has not been validated. Third, each student participated to a maximum of three time points and most of them participated only in one or two time points. Further studies with less incomplete data across time points are warranted to confirm the accuracy of the full trajectories' estimation, ascertain the precision of parameter estimates, and avoid false-negative results. Fourth, interactions between curriculum year and biopsychosocial

covariates were not examined to avoid an increase in Type I error. Future studies focusing on the timely impact of specific covariates are needed to understand how they potentially exacerbate or alleviate medical students' mental health and burnout across time. Finally, the study was run in one single Swiss university, which limits the generalizability of the findings. It also made it impossible to evaluate the impact of university-specific factors such as the pedagogical approach or the study program. Replication in other universities and countries is needed to corroborate this study's results and explore university-related covariates of medical students' mental health and burnout.

## Conclusion

This paper presents one of the rare longitudinal studies of mental health and burnout spanning from the first to the last year of medical school. Results indicates that an important minority of students present mental health issues or burnout at some point during their medical curriculum. Longitudinal analyses show that medical students' mental health improves during medical school in terms of depression symptoms, suicidal ideation, and stress, although suicidal ideation presents re-increase in the last curriculum years. For burnout, we see a worsening along the curriculum years with more cynicism, less academic efficacy, and peeks of emotional exhaustion in the second and last years of medical school. Therefore, a non-negligible proportion of residents enter their professional life with mental health issues or burnout that not only negatively impact their life, but might also hinder their professionalism [9,48] and in turn affect their medical practice. The poor quality of life of young physicians is well documented [49], maybe the root of this malaise can be found in the last years of the medical studies as suggested by our results. The present study indicates coping strategies, social support, and satisfaction with health as the most influential covariates that should be primarily targeted by interventions aiming to strengthen medical students' resources in the face of mental health issues and burnout. Nevertheless, more research is needed on the stressors experienced by medical students in the beginning versus in the end of the medical curriculum to confirm their differential impacts on mental health and burnout and enable to draw more tailored avenues for intervention and improve the curriculum.

## Supporting information

**S1 Fig. Participation flow chart.**
(PDF)

**S1 Table. Correlations between the variables of interest (N = 1595).**
(PDF)

**S1 File. Equations for the linear mixed model analyses.**
(PDF)

## Acknowledgments

The authors want to thank Sylvie Felix and Fabienne Thévenaz for their help in the ETMED-L project's data collection.

## Author Contributions

**Conceptualization:** Valerie Carrard, Sylvie Berney, Céline Bourquin, Marianne Schmid Mast, Alexandre Berney.

**Data curation:** Valerie Carrard, Enrique Castelao.

**Formal analysis:** Valerie Carrard, Setareh Ranjbar.

**Funding acquisition:** Céline Bourquin, Alexandre Berney.

**Investigation:** Valerie Carrard, Sylvie Berney, Céline Bourquin, Alexandre Berney.

**Methodology:** Valerie Carrard, Céline Bourquin, Setareh Ranjbar, Enrique Castelao, Martin Preisig, Alexandre Berney.

**Project administration:** Valerie Carrard, Céline Bourquin, Alexandre Berney.

**Resources:** Martin Preisig.

**Supervision:** Céline Bourquin, Alexandre Berney.

**Writing – original draft:** Valerie Carrard.

**Writing – review & editing:** Valerie Carrard, Sylvie Berney, Céline Bourquin, Setareh Ranjbar, Enrique Castelao, Katja Schlegel, Jacques Gaume, Pierre-Alexandre Bart, Marianne Schmid Mast, Martin Preisig, Alexandre Berney.

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
