## [Decision Letter · Decision Letter 0]

18 Jan 2024

PONE-D-23-36664Mental health and burnout during medical school: Longitudinal evolution and covariatesPLOS ONE

Dear Dr. Berney,

Thank you for submitting your manuscript to PLOS ONE. After careful consideration, we feel that it has merit but does not fully meet PLOS ONE’s publication criteria as it currently stands. Therefore, we invite you to submit a revised version of the manuscript that addresses the points raised during the review process.

**ACADEMIC EDITOR:**

Please take into account the suggestions made by the two reviewers.

We look forward to receiving your revised manuscript.

Kind regards,

Santiago Gascón, PhD

Academic Editor

PLOS ONE

Journal Requirements:

"This work was supported by the Swiss National Science Foundation (grant number 10001C_197442). The CoLaus|PsyCoLaus offspring study was supported by unrestricted research grants from GlaxoSmithKline, the Faculty of Biology and Medicine of Lausanne, the Swiss National Science Foundation (grants 3200B0–105993, 3200B0-118308, 33CSCO-122661, 33CS30-139468, 33CS30-148401, 33CS30_177535, 3247730_204523, 324730_189130) and the Swiss Personalized Health Network (grant 2018DRI01)."

Additional Editor Comments:

Dear author,

Thank you very much for submitting your article to PLOS ONE.

Please take into account the suggestions made by the two independent reviewers on your article.

Thank you very much.

Reviewers' comments:

Reviewer's Responses to Questions

**Comments to the Author**

1. Is the manuscript technically sound, and do the data support the conclusions?

Reviewer #1: Yes

Reviewer #2: Yes

2. Has the statistical analysis been performed appropriately and rigorously? 

Reviewer #1: Yes

Reviewer #2: Yes

3. Have the authors made all data underlying the findings in their manuscript fully available?

Reviewer #1: Yes

Reviewer #2: No

4. Is the manuscript presented in an intelligible fashion and written in standard English?

Reviewer #1: Yes

Reviewer #2: Yes

5. Review Comments to the Author

Reviewer #1: Overall, the manuscript provides a great depth of understanding of mental health-related stressors among a critical group (medical students). The manuscript is timely and contributes greatly to the literature to ensure that a conducive training experience is created for such an important workforce. I believe the paper can be strengthened in the following areas:

1. For anonymity purposes, it will be good to black out the name of the institution. I would say, “Data was collected from a medical school in Switzerland”.

2. Page 5 line 104. There is a typographical error that needs to be addressed.

3. In the Maslach Burnout Inventory Student Survey, does Academic Efficacy address the personal accomplishment component of the inventory? If yes, it will be good to clarify that adaptation.

4. For the social support variable, do you check for internal consistency? If you do, please report Cronbach's alpha. Also, were these questions that the authors created to measure that construct, or a validated measure was used? It will be good to clarify that. Please provide the sample size in the descriptive table as well.

5. Page 5 line 142. There is a typographical error that needs to be addressed.

6. Authors indicate that they replaced all missing values by mean scores if missingness was less than 20%. I believe the authors used the mean imputation method. Considering the controversies around the different imputation methods it would be great to provide some citations/justifications. Taking out items with more than 20% missingness seems concerning. I believe a citation can be provided to explain the approach taken. This should be enough. Alternatively, authors can also explore some imputation methods (Eg., Nearest Neighbor Imputation, or MLE if the type of missingness is clear to the Authors).

7. I appreciate the explanation of the model-building process. However, I believe the readers will benefit from step-by-step guidance on the model-building process with relevant equations attached as well as some explanations of the reported ICC values.

8. Authors report that “medical students’ mental health gradually improves in terms of depression symptoms, suicidal ideation, and stress, although suicidal ideation increases again during the last year of medical school and anxiety symptoms remain stable throughout the curriculum”. This is a profound and interesting finding. Could Authors provide some theoretical explanation as to why they think this is happening? Theories are always great!

Reviewer #2: The study is very interesting, yet some improvements are needed:

1. provide clearer explanations for the observed temporal trends in mental health and burnout. Highlight the implications of these trends for medical students, and discuss how they align or contrast with previous literature.

2. Given the sensitivity of the topic, provide a more in-depth discussion on the increase in suicidal ideation during the last year of medical school. Explore potential contributing factors, and discuss the importance of targeted interventions for this specific period.

3. While the study mentions influential biopsychosocial covariates, a more detailed discussion on how these factors interact and potentially exacerbate or alleviate mental health issues and burnout would enhance the manuscript. Consider delving into the practical implications of these findings for support programs.

4. Given the gender differences observed, discuss the potential reasons behind these variations and their implications for interventions. If applicable, explore whether specific support mechanisms are needed for different genders.

5. Clearly outline the practical implications of the study findings for medical schools and institutions. Discuss how the identified protective factors (e.g., coping strategies, social support) can be integrated into existing support systems for medical students.

6. Strengthen the conclusion by summarizing key findings concisely and reiterating the practical implications. Discuss potential future research directions based on the gaps identified in the study.

6. PLOS authors have the option to publish the peer review history of their article (what does this mean?). If published, this will include your full peer review and any attached files.

Reviewer #1: No

Reviewer #2: No

---

## [Author Response · Author response to Decision Letter 0]

29 Feb 2024

Dear Dr Santiago Gascón,

Thank you for giving us the opportunity to revise and resubmit our manuscript for publication consideration in PLOS ONE. We carefully addressed each of the editors’ and reviewers’ comments as detailed in the following. We hope that the changes made to the manuscript will be to your satisfaction. 

Journal Requirements:

-The manuscript and supporting information files were revised to meet PLOS ONE style. Note that track change was not used for these specific modifications for readability’s sake.

-We totally agree that open data availability should be ascertained as often as possible. That is why we openly share the data used in our main analyses (N=1595) in a public repository (Zenodo), with a doi that is now activated (http://doi.org/10.5281/zenodo.8405764) and indicated in the “Data Review URL”. However, we compared our openly available data to a third-party general population dataset (N=91) that we are not allowed to share as it contains potentially sensitive personal information on participants. As indicated in our data availability statement: 

“The data that support the main findings of this study are openly available in zenodo at http://doi.org/10.5281/zenodo.8405764. For the comparison analyses, the data of CoLaus|PsyCoLaus study used in this article cannot be fully shared as they contain potentially sensitive personal information on participants. According to the Ethics Committee for Research of the Canton of Vaud, sharing these data would be a violation of the Swiss legislation with respect to privacy protection. However, coded individual-level data that do not allow researchers to identify participants are available upon request to researchers who meet the criteria for data sharing of the CoLaus|PsyCoLaus Datacenter (CHUV, Lausanne, Switzerland). Any researcher affiliated to a public or private research institution who complies with the CoLaus|PsyCoLaus standards can submit a research application to research.colaus@chuv.ch or research.psycolaus@chuv.ch. Proposals requiring baseline data only, will be evaluated by the baseline (local) Scientific Committee (SC) of the CoLaus and PsyCoLaus studies. Proposals requiring follow-up data will be evaluated by the follow-up (multicentric) SC of the CoLaus|PsyCoLaus cohort study. Detailed instructions for gaining access to the CoLaus|PsyCoLaus data used in this study are available at www.colaus-psycolaus.ch/professionals/how-to-collaborate.”

"This work was supported by the Swiss National Science Foundation (grant number 10001C_197442). The CoLaus|PsyCoLaus offspring study was supported by unrestricted research grants from GlaxoSmithKline, the Faculty of Biology and Medicine of Lausanne, the Swiss National Science Foundation (grants 3200B0–105993, 3200B0-118308, 33CSCO-122661, 33CS30-139468, 33CS30-148401, 33CS30_177535, 3247730_204523, 324730_189130) and the Swiss Personalized Health Network (grant 2018DRI01)."

-The funders had no role in the study. Thus, as also indicated in our Cover Letter, our funding statement should read as follow: "This work was supported by the Swiss National Science Foundation (grant number 10001C_197442). The CoLaus|PsyCoLaus offspring study was supported by unrestricted research grants from GlaxoSmithKline, the Faculty of Biology and Medicine of Lausanne, the Swiss National Science Foundation (grants 3200B0–105993, 3200B0-118308, 33CSCO-122661, 33CS30-139468, 33CS30-148401, 33CS30_177535, 3247730_204523, 324730_189130) and the Swiss Personalized Health Network (grant 2018DRI01). The funders had no role in study design, data collection and analysis, decision to publish, or preparation of the manuscript."

-The major part of our data (N=1595) is openly available in a public repository (Zenodo), with the a doi that is now activated (http://doi.org/10.5281/zenodo.8405764) and indicated in the “Data Review URL”. However, a minor part of the data used in our analyses (N=91) cannot be openly shared as indicated in our Data Availability Statement (see also our answer to the journal requirements no 2).

Reviewers' comments:

Reviewer #1: 

Overall, the manuscript provides a great depth of understanding of mental health-related stressors among a critical group (medical students). The manuscript is timely and contributes greatly to the literature to ensure that a conducive training experience is created for such an important workforce. I believe the paper can be strengthened in the following areas:

-We thank Reviewer#1 for their encouraging words and comments that helped us improve our manuscript. 

1. For anonymity purposes, it will be good to black out the name of the institution. I would say, “Data was collected from a medical school in Switzerland”.

-Thank you for this suggestion. However, blackening out the institution name would not prevent identification of the institution given that we refer to the published project protocol that indicates the institution name. Also, the authors’ affiliation indicates rather unequivocally where the data were collected. The ETMED-L project strives for the transparency of data, and we would thus prefer to indicate the name of the institution.

2. Page 5 line 104. There is a typographical error that needs to be addressed.

-Thank you for your keen eye. The typo has been corrected. 

3. In the Maslach Burnout Inventory Student Survey, does Academic Efficacy address the personal accomplishment component of the inventory? If yes, it will be good to clarify that adaptation.

-Yes, the Academic efficacy of the student version of the Maslach Burnout Inventory indeed corresponds to the personal accomplishment component of the original inventory [1]. This has been clarified in the manuscript as follows:

“For burnout, the Maslach Burnout Inventory Student-Survey [MBI-SS; 17] was used. The Maslach Burnout Inventory is the most widely used measure of burnout that assesses three dimensions: emotional exhaustion, depersonalization, and personal accomplishment (reversed dimension). The students’ version used in the present study has been validated among students and measures the corresponding three dimensions of Emotional Exhaustion, Cynicism, and Academic Efficacy (reversed dimension).”

4. For the social support variable, do you check for internal consistency? If you do, please report Cronbach's alpha. Also, were these questions that the authors created to measure that construct, or a validated measure was used? It will be good to clarify that. Please provide the sample size in the descriptive table as well.

-Thank you for your suggestions. The description of the social support variable has been revised to indicate the Cronbach’s alpha and that the measure was adapted from a larger cohort study’s questionnaire. 

“Social support was measured as the average score of two questions adapted from a larger cohort study’s questionnaire [18]: “If necessary, in your opinion, to what extent can someone provide you with practical help, this means concrete help or useful advice, if 0 means "not at all" and 10 "a great deal"?” for practical support and “To what extent can someone be available in case of need and show understanding, by talking with you for example, if 0 means "not at all" and 10 "a great deal"?” for emotional support (Cronbach’s alpha = .73).”

Moreover, as requested, the sample sizes were added in Table 1. 

5. Page 5 line 142. There is a typographical error that needs to be addressed.

-The typo has been corrected. 

6. Authors indicate that they replaced all missing values by mean scores if missingness was less than 20%. I believe the authors used the mean imputation method. Considering the controversies around the different imputation methods it would be great to provide some citations/justifications. Taking out items with more than 20% missingness seems concerning. I believe a citation can be provided to explain the approach taken. This should be enough. Alternatively, authors can also explore some imputation methods (Eg., Nearest Neighbor Imputation, or MLE if the type of missingness is clear to the Authors).

-Thank you for this comment. It is true that imputation methods such as nearest neighbor or random forest are generally more advisable than mean scores. However, Lee and colleagues [3] showed that when the missing rates are very low (i.e. lower than 5%), it is unlikely to be much gain from multiple imputation. Given that our highest missing rate at the item level was 0.97% (coping and social support items), multiple imputation was deemed unnecessary and the dismissing of scores based on less than 20% of the missing was considered as an unsubstantial loss of information. The description of the missing data approach has been modified to clarify this point:

“The highest missing rate at the item level was 0.97% (coping and social support items). It has been shown that it is unlikely to be much gain from multiple imputation when missing rates are lower than 5% [25]. Thus, missing data at the item level were replaced by mean scores if less than 20% of the items were missing. If 20% or more of the items were missing, the total score was considered missing and missing data at the score level were then handled with full information maximum likelihood in the LMMs.”

7. I appreciate the explanation of the model-building process. However, I believe the readers will benefit from step-by-step guidance on the model-building process with relevant equations attached as well as some explanations of the reported ICC values

-Thank you for this suggestion. As recommended, the Statistical analysis section has been revised as follows to offer a step-by-step guidance and clarify the reported indices. Additionally, the related equations are now available as supporting information to the manuscript. 

“The LMMs analyses followed the subsequent steps for each mental health and burnout indicators separately (see S1 File for the related equations). In a first step, Model 1 was fitted considering the student specific effects as random intercepts to account for the clustering in the data due to the repeated measures for each medical student and including all the biopsychosocial covariates described in the Measure section. In a second step, two different temporal variance-covariance structures were fitted to potentially account for the temporal spillover of mental health and burnout (i.e. the influence of past observation on current observation). The same model as in the first step was fitted, where an Autoregressive Covariance Structure of AR(1) was considered in Model 2 and an Autoregressive/Moving Average Covariance Structure of ARMA(1,1) in Model 3. Then, the best fitting model between 1, 2, and 3 was selected using likelihood ratio tests. In a third step, the non-linear evolution of mental health and burnout was tested by including the time as quadratic in Model 4 and cubic in Model 5 with the temporal structure selected in the second step. Using likelihood ratio tests, Models 4 and 5 were then compared to the linear Model 1, 2, or 3 according to the temporal structure selected in the second step. The best fitting model of this third step was presented as the final model. Note that Restricted Maximum Likelihood method was used to produce unbiased estimates of variance and covariance parameters [23,24]. For all the final models, we report marginal R2 as the percentage of variation explained by the fixed part of the model, conditional R2 as the percentage of variation explained by both fixed and random part of the model, and Intraclass Correlation Coefficient (ICC) as the proportion of total between-student variance.”

8. Authors report that “medical students’ mental health gradually improves in terms of depression symptoms, suicidal ideation, and stress, although suicidal ideation increases again during the last year of medical school and anxiety symptoms remain stable throughout the curriculum”. This is a profound and interesting finding. Could Authors provide some theoretical explanation as to why they think this is happening? Theories are always great!

-We believe that differences between the stressors faced by medical students in the beginning of their studies and those faced at the end of the curriculum can partly explain our results. Indeed, as described in our Discussion section, the stressors met at the beginning of the studies might be the one that students from other faculties also meet, whereas those in the end of medical school might be more specific to medical students with the first entry into clinical practice and the perspective of the transition to residency. The Stress and Coping Model of Lazarus and Folkman [8] could explain the lessening impact of the stressors encountered in the beginning of the studies on mental health as students progressively adapt to their new lifestyle by modifying their appraisal of the situation and developing new coping strategies. On the other hand, the job Demand Control Model [9] could explain the observed increase in burnout in the end of the curriculum that is characterized by especially high demands and low sense of control. The following paragraph has been revised in order to present the theoretical model supporting the interpretation of our results: 

“Indeed, the beginning of medical school is characterized by difficulties that might be faced by any new university students, such as change of social environment, adaptation to a new learning style, high study workload, strong competition, and stressful examinations. The impact of these stressors is likely to lessen along the curriculum years as students adapt to what has become their new lifestyle. As proposed by the Stress and Coping Model [30], medical students’ mental health might improve as they progressively appraise their new situation as less challenging and acquire new coping strategies to manage the stressors encountered. However, the end of medical school can bring new types of stressors that might be more specific to medical students. Indeed, through clerkships, medical students are gradually exposed to their potential future practice. They experience the relationship with patients, but also with co-workers and their hierarchy and might face difficult situations where they are confronted with the complexity, feeling of powerlessness, and lack of control that can characterize medical practice. Based on the Job Demand Control Model (JDCM [31]), such an increase in job demands (e.g., more responsibilities and workload [32]) coupled with perceived loss of control (e.g., role conflicts and limit of their theoretical knowledge [32]) can lead to increased stress and strain reactions, including burnout.”

Reviewer #2:

The study is very interesting, yet some improvements are needed:

1. provide clearer explanations for the observed temporal trends in mental health and burnout. Highlight the implications of these trends for medical students, and discuss how they align or contrast with previous literature.

-Thank you for this suggestion. Reviewer 1 also points out the need for further explanation of the observed trajectories. This is a very good point that we addressed by proposing theoretical models that can shed light on the temporal changes observed and their implications for students (see also our answer to Reviewer 1’s comment no 8). The trends observed in mental health rather align with past literature, whereas the alignment with past literature of the results pertaining to burnout is difficult to assess due to the lack of longitudinal studies spanning the entire course of medical studies. The third paragraph of our Discussion section has been adapted to better underline the comparison to past studies in the field. 

2. Given the sensitivity of the topic, provide a more in-depth discussion on the increase in suicidal ideation during the last year of medical school. Explore potential contributing factors, and discuss the importance of targeted interventions for this specific period.

-Medical students’ suicidal ideation is indeed a concerning matter. However, our results do not indicate the last years of the curriculum as the most concerning period. Indeed, suicidal ideations’ means are higher in the year 2, 3, and 4 than in year 5 and 6 (see Table 2). Moreover, our study was not meant and is not built to thoroughly examine the suicide crisis factors. Thus, further studies specifically focused on suicidality are needed to shed light on the specific causes of suicidal ideations and confidently propose targeted interventions. The present study does not indicate any factor influencing solely suicidal ideation. Thus, the interventions proposed here target more generally mental health issues as well as burnout. Nonetheless, as recommended, we enriched our description of the practical implications for support services to offer to medical students as follows: 

“In our university, numerous extra-curricular activities are proposed by different associations to bring students together and thus increase their social support resources. The problem that might face medical students especially is the difficulty to find room in their busy schedule for such activities, which clearly questions the structure of the curriculum. In addition, medical students are somewhat isolated from the rest of the university campus. Other services available in our university are psychotherapeutic consultations on campus. However, these structures are still limited in their means in comparison to the needs and our results pleads for an increase in the dedicated resources. Emotion-focused coping, defined as the detrimental use of rumination, negative emotions, denial, and giving up, is the most influential among the covariates tested. It thus seems essential to warrant accessible psychotherapeutic help, which has the potential to impact medical students’ coping strategies and in turn their mental health [46,47].”

3. While the study mentions influential biopsychosocial covariates, a more detailed discussion on how these factors interact and potentially exacerbate or alleviate mental health issues and burnout would enhance the manuscript. Consider delving into the practical implications of these findings for support programs.

-Thank you for this comment. Our study includes a total of 12 biopsychosocial covariates and 7 different indicators of mental health and burnout. Testing the time interactions of each covariate for each indicator would result in the fitting of 84 supplementary models. We considered that such a high number of models would preclude the clarity of the manuscript and add too much risk of Type I error. We added this as a limitation of our study and call for future studies specifically focusing on how the covariates interact with time and potentially exacerbate or alleviate mental health issues and burnout: 

“Fourth, interactions between curriculum year and biopsychosocial covariates were not examined to avoid an increase in Type I error. Future studies focusing on the timely impact of specific covariates are needed to understand how they potentially exacerbate or alleviate medical students’ mental health and burnout across time.”

4. Given the gender differences observed, discuss the potential reasons behind these variations and their implications for interventions. If applicable, explore whether specific support mechanisms are needed for different genders.

-Thank you for this suggestion. Gender differences are now discussed as follows:

“The present study replicates most of the gender differences usually observed in the general population [36] with students identifying as male presenting less mental health issues than students identifying as female or non-binary. For burnout, our study also aligns with past literature showing inconsistent gender differences [37–40]. We indeed found that students identifying as male present less emotional exhaustion, but more cynicism, and less academic efficacy (reversed dimension of burnout) than students identifying as female or non-binary. Gender differences in mental health and wellbeing have been attributed to multiple factors including differences between women and men in the stressors faced, the coping strategies used, and developed vulnerabilities [36]. Indeed, the few studies testing gender differences in the stressors faced by medical students indicate that female medical students are more affected by specific stressors such as the heavy workload of the studies [41], the interactions with teaching staff [42], or the exposure to human suffering [43]. Regarding coping resources, studies showed that female medical students use significantly more emotional support and instrumental support than male students, whereas male students use significantly more humor as coping mechanism [44]. More studies on the gender differences in medical students’ stressors and resources and their impact are needed to potentially propose avenues for gender-specific interventions.” 

5. Clearly outline the practical implications of the study findings for medical schools and institutions. Discuss how the identified protective factors (e.g., coping strategies, social support) can be integrated into existing support systems for medical students.

-As recommended, we enriched our description of support services having the potential to influence coping strategies and social support (see also our response to comment no 2).

6. Strengthen the conclusion by summarizing key findings concisely and reiterating the practical implications. Discuss potential future research directions based on the gaps identified in the study.

-Thank you for this suggestion, the conclusion section has been revised as suggested and now reads as follow:

“This paper presents one of the rare longitudinal studies of mental health and burnout spanning from the first to the last year of medical school. Results indicates that an important minority of students present mental health issues or burnout at some point during their medical curriculum. Longitudinal analyses show that medical students’ mental health improves during medical school in terms of depression symptoms, suicidal ideation, and stress, although suicidal ideation presents re-increase in the last curriculum years. For burnout, we see a worsening along the curriculum years with more cynicism, less academic efficacy, and peeks of emotional exhaustion in the second and last years of medical school. Therefore, a non-negligible proportion of residents enter their professional life with mental health issues or burnout that not only negatively impact their life, but might also hinder their professionalism [9,48] and in turn affect their medical practice. The poor quality of life of young physicians is well documented [49], maybe the root of this malaise can be found in the last years of the medical studies as suggested by our results. The present study indicates coping strategies, social support, and satisfaction with health as the most influential covariates that should be primarily targeted by interventions aiming to strengthen medical students’ resources in the face of mental health issues and burnout. Nevertheless, more research is needed on the stressors experienced by medical students in the beginning versus in the end of the medical curriculum to confirm their differential impacts on mental health and burnout and enable to draw more tailored avenues for intervention and improve the curriculum.”

References: 

1. Schaufeli WB, Martínez IM, Pinto AM, Salanova M, Bakker AB. Burnout and engagement in university students: a cross-national study. J. Cross-Cult. Psychol. 2002;464–81. 

2. Voorpostel M, Tillmann R, Lebert F, Kuhn U, Lipps O, Ryser VA, et al. Swiss Household Panel Userguide (1999-2020), Wave 22. Lausanne: FORS; 2022. 

3. Lee KJ, Roberts G, Doyle LW, Anderson PJ, Carlin JB. Multiple imputation for missing data in a longitudinal cohort study: A tutorial based on a detailed case study involving imputation of missing outcome data. Int. J. Soc. Res. Methodol. 2016;19:575–91. 

4. StataCorp. Stata Statistical Software: Release 17. StataCorp LLC. College Station, TX: 2021. 

5. R Core Team. R: A language and environment for statistical computing. R Foundation for Statistical Computing. Vienna, Austria: 2022. 

6. Pinheiro J, Bates D. Mixed-Effects Models in S and S-PLUS. Springer Science & Business Media; 2006. 

7. Gałecki A, Burzykowski T. Linear Mixed-Effects Model. In: Gałecki A, Burzykowski T, editors. Linear Mixed-Effects Models Using R: A Step-by-Step Approach. New York, NY: Springer; 2013. page 245–73.

8. Lazarus RS, Folkman S. Stress, appraisal, and coping. Springer Publishing Company; 1984. 

9. Karasek RA. Job demands, job decision latitude, and mental strain: Implications for job redesign. Adm. Sci. Q. 1979;24:285–308. 

10. Moczko TR, Bugaj TJ, Herzog W, Nikendei C. Perceived stress at transition to workplace: a qualitative interview study exploring final-year medical students’ needs. Adv. Med. Educ. Pract. 2016;7:15–27. 

11. Yusoff MSB. Interventions on medical students’ psychological health: A meta-analysis. J. Taibah Univ. Med. Sci. 2014;9:1–13. 

12. Huang J, Nigatu YT, Smail-Crevier R, Zhang X, Wang J. Interventions for common mental health problems among university and college students: A systematic review and meta-analysis of randomized controlled trials. J. Psychiatr. Res. 2018;107:1–10. 

13. Rosenfield S, Mouzon D. Gender and mental health. In: Aneshensel CS, Phelan JC, Bierman A, editors. Handbook of the Sociology of Mental Health. Dordrecht: Springer Netherlands; 2013. page 277–96.

14. Li Y, Cao L, Mo C, Tan D, Mai T, Zhang Z. Prevalence of burnout in medical students in China. Medicine (Baltimore) 2021;100:e26329. 

15. Frajerman A, Morvan Y, Krebs MO, Gorwood P, Chaumette B. Burnout in medical students before residency: A systematic review and meta-analysis. Eur. Psychiatry 2019;55:36–42. 

16. Almutairi H, Alsubaiei A, Abduljawad S, Alshatti A, Fekih-Romdhane F, Husni M, et al. Prevalence of burnout in medical students: A systematic review and meta-analysis. Int. J. Soc. Psychiatry 2022;68:1157–70. 

17. Pacheco JP, Giacomin HT, Tam WW, Ribeiro TB, Arab C, Bezerra IM, et al. Mental health problems among medical students in Brazil: a systematic review and meta-analysis. Braz. J. Psychiatry 2017;39:369–78. 

18. Toews JA, Lockyer JM, Dobson DJ, Simpson E, Brownell AK, Brenneis F, et al. Analysis of stress levels among medical students, residents, and graduate students at four Canadian schools of medicine. Acad. Med. 1997;72:997–1002. 

19. Backović DV, Ilić Živojinović J, Maksimović J, Maksimović M. Gender differences in academic stress and burnout among medical students in final years of education. Psychiatr. Danub. 24:175–81. 

20. Pitkala KH, Mantyranta T. Professional socialization revised: medical students’ own conceptions related to adoption of the future physician’s role–a qualitative study. Med. Teach. 2003;25:155–60. 

21. Spataro BM, Tilstra SA, Rubio DM, McNeil MA. The Toxicity of Self-Blame: Sex Differences in Burnout and Coping in Internal Medicine Trainees. J. Womens Health 2016;25:1147–52. 

22. Dyrbye LN, Thomas MR, Shanafelt TD. Systematic review of depression, anxiety, and other indicators of psychological distress among U.S. and Canadian medical students. Acad. Med. 2006;81:354–73. 

23. Dyrbye LN, Massie FS, Eacker A, Harper W, Power D, Durning SJ, et al. Relationship between burnout and professional conduct and attitudes among US medical students. JAMA 2010;304:1173–80. 

24. Raj KS. Well-being in residency: A systematic review. J. Grad. Med. Educ. 2016;8:674–84.

---

## [Editor Report · Decision Letter 1]

28 Mar 2024

Mental health and burnout during medical school: Longitudinal evolution and covariates

PONE-D-23-36664R1

Dear Dr. Berney,

We’re pleased to inform you that your manuscript has been judged scientifically suitable for publication and will be formally accepted for publication once it meets all outstanding technical requirements.

Kind regards,

Santiago Gascón, PhD

Academic Editor

PLOS ONE

Additional Editor Comments (optional):

The authors of the manuscript entitled “Mental health and burnout during medical school: Longitudinal evolution and covariates” have answered the questions raised by the various reviewers correctly in my view. Una vez leído el manuscrito final parece mucho más claro, especialmente las secciones de Método, de Discusión y Conclusiones.

I therefore recommend its publication in PLOS ONE.

---

## [Editor Report · Acceptance letter]

4 Apr 2024

PONE-D-23-36664R1 

PLOS ONE

Dear Dr. Berney, 

I'm pleased to inform you that your manuscript has been deemed suitable for publication in PLOS ONE. Congratulations! Your manuscript is now being handed over to our production team.

Kind regards, 

on behalf of

Dr. Santiago Gascón 

Academic Editor

PLOS ONE